# Performance Optimization and Exergy Analysis of Thermoelectric Heat Recovery System for Gas Turbine Power Plants

**DOI:** 10.3390/e25121583

**Published:** 2023-11-25

**Authors:** Ahmad M. Alsaghir, Je-Hyeong Bahk

**Affiliations:** Department of Mechanical and Materials Engineering, University of Cincinnati, Cincinnati, OH 45221, USA; alsagham@mail.uc.edu

**Keywords:** thermoelectric, waste heat recovery, exergy analysis, gas turbine, energy efficiency

## Abstract

Thermoelectric (TE) waste heat recovery has attracted significant attention over the past decades, owing to its direct heat-to-electricity conversion capability and reliable operation. However, methods for application-specific, system-level TE design have not been thoroughly investigated. This work provides detailed design optimization strategies and exergy analysis for TE waste heat recovery systems. To this end, we propose the use of TE system equipped on the exhaust of a gas turbine power plant for exhaust waste heat recovery and use it as a case study. A numerical tool has been developed to solve the coupled charge and heat current equations with temperature-dependent material properties and convective heat transfer at the interfaces with the exhaust gases at the hot side and with the ambient air at the heat sink side. Our calculations show that at the optimum design with 50% fill factor and 6 mm leg thickness made of state-of-the-art Bi_2_Te_3_ alloys, the proposed system can reach power output of 10.5 kW for the TE system attached on a 2 m-long, 0.5 × 0.5 m^2^-area exhaust duct with system efficiency of 5% and material cost per power of 0.23 $/W. Our extensive exergy analysis reveals that only 1% of the exergy content of the exhaust gas is exploited in this heat recovery process and the exergy efficiency of the TE system can reach 8% with improvement potential of 85%.

## 1. Introduction

Increasing efficiency of an energy conversion system is of great importance in our society as we tackle the ongoing climate crisis and aim to reduce the use of fossil fuel resources [1]. About 72% of the global energy consumption is wasted after the conversion processes [2]. Moreover, the recent U.S. energy chart shows that about 60% of the energy generated is rejected to the environment mostly as waste heat [3]. From this perspective, waste heat recovery (WHR) is one of the promising approaches that can increase the efficiency of energy conversion processes and reduce the use of fossil fuels. Waste heat recovery systems can be categorized into three groups: Heat-to-Heat, Heat-to-Work, and Heat-to-Power [4]. This study investigates the utilization of the Heat-to-Power WHR approach using solid-state thermoelectric generators (TEGs) in a gas turbine power plant as a case study as schematically shown in Figure 1.

Gas turbine cycles are one of the most popular energy conversion cycles that convert the chemical energy of a fuel to mechanical energy to rotate turbine blades, which in turn generates electricity. As schematically shown in Figure 1, a typical gas turbine power plant comprises four main components: a compressor to pressurize the input gas, a combustion chamber to add energy to the compressed gas, a prime mover (turbine) to extract the energy from the hot gas, and an exhaust system to capture the toxic combustion products before releasing the flue gases to the atmosphere [5]. Despite its relatively high energy efficiency, a gas turbine wastes a vast amount of energy by releasing high temperature gases into the environment. The typical temperature of the exhaust gases from a simple gas turbine cycle is in the range of 400–500 °C [6].

There are two known approaches to recover waste heat from a gas turbine: regeneration and cogeneration. A regenerator (or recuperator) is a heat exchanger used to preheat the compressed gas with the exhaust gas before it enters the combustion chamber, thereby decreasing the heat input requirement by recycling waste heat from the exhaust [5]. However, in high-pressure-ratio compressors, the temperature of the compressed gases can be already higher than the temperature of the exhaust gases, so a regenerator is unviable in such a case as it will decrease the compressed gas temperature. Also, the pressure drop across the regenerator can adversely increase the power consumption of the cycle and thus reduce the overall efficiency. Hence, regenerative gas turbines are relatively scarce [7]. On the other hand, cogeneration is the process of producing multiple useful forms of energy from a single energy source [8]. For gas turbines, the released gases can be used to generate superheated vapor for a subsequent Rankine cycle in combined cycles or can be exploited in various processes where heat is the main input such as in chemical, paper, or food processing industry, and also in air conditioning, desalination, etc. [9]. However, the high capital cost and special installation requirements associated with these cogeneration systems could hinder their utilization. Furthermore, gas turbines are often desired to operate on part load, which reduces the amount of released heat and makes it less attractive for use in many of the aforementioned systems.

Over the past decades, thermoelectric energy conversion has attracted great attention as a heat recovery technology because of its capability to directly convert waste heat to electrical power based on the Seebeck effect without moving parts and fluids involved [10]. A thermoelectric generator (TEG) is a solid-state device comprised of multiple n-type and p-type semiconductor elements or TE legs that are connected electrically in series and thermally in parallel as shown in Figure 1a. When heat flows through the module, a voltage is induced in each of the TE legs by the Seebeck effect, which delivers power to the load. TE system offers small form factors, no moving parts, robust operation, and minimal impacts to the application system, e.g., gas turbine.

Maximum power output or efficiency is achieved when the module design parameters such as fill factor (fractional area coverage by TE legs) and leg thickness are optimized for simultaneous electrical and thermal load matching against the external electrical and thermal resistances [11]. Hence, custom design optimization of TE system is necessary for a specific application system as the conditions of external resistances vary depending on the application. The efficiency of a thermoelectric material is largely determined by the dimensionless figure-of-merit *ZT* defined as
(1)$$ZT=\frac{S^{2}\sigma }{\kappa }T$$
where *S* is the Seebeck coefficient or thermopower, σ is the electrical conductivity, κ is the thermal conductivity, and *T* is the absolute temperature [12]. Equation (1) reveals that a good thermoelectric material is characterized by high power factor ($S^{2}\sigma )$ and low thermal conductivity. Researchers have shown that this could be attained by nanostructuring [13]. Nanostructuring of thermoelectric materials is a promising technique that can suppress the phonon transport to reduce thermal conductivity, while maintaining the electron transport or the power factor high to improve the *ZT*. For example, nanostructured Bi_2_Te_3_ alloys enhanced the *ZT* values of the n-type and the p-type to ~1.4 and ~1.0, respectively [14,15]. Moreover, the rhombohedral complex molecular structures of GeTe and SnSe have made them excellent thermoelectric materials with *ZT* above 2.0 in the mid-temperature range [16,17]. There are several review papers that describe the recent advancement of thermoelectric materials [13,18,19].

Recently, there has been a great deal of research on TE waste heat recovery systems. Kuroki et al. [20] demonstrated the feasibility of using TEGs to recover waste heat generated from a steel making process. The system consisted of 16 TEG modules of 5 × 5 cm^2^ mounted above the radiating slab; each module produced 18 W under the temperature difference of 220 K. The maximum power density of 1.32 kW/m^2^ was achieved in this study when the hot side temperature was 1188 K. Referring to this study, Ghosh et al. [21] provided a detailed theoretical investigation to further optimize the module performance. Yazawa et al. [22] exploited the large temperature difference between the burner flame and the pressurized steam in a steam turbine cycle to produce additional electrical power and enhance the overall efficiency using TEGs as a topping cycle. They carried out an optimization study on the interface temperature between the TEG and the steam pipes for better energy economy. A follow-up study was carried out by the same team [23] to investigate the efficiency enhancement of advanced supercritical steam turbines as a result of adding the TE topping generators. Hasani et al. [24] experimentally investigated a proton-exchange membrane (PEM) fuel cell thermoelectric heat recovery system. They reported that when TEGs were integrated with a 5 kW PEM fuel cell to recover waste heat, they could produce up to 800 W more electric power with TE modules. However, the TE efficiency was only around 0.35% at hot water temperature of 68 °C due to the relatively small temperature difference applied. Lai et al. [25] investigated the viability of performance enhancement for a dye-sensitized solar cell coupled with selective solar absorber and flexible annular thermoelectric generator. The results revealed that the proposed system could produce 10.5% higher power density and 39% more maximum energy. In addition to these studies, there have been several studies on TE waste recovery in cement manufacturing processes [26,27,28,29,30] as well. In this process, a vast amount of heat is wasted from the shell of the rotating kiln. Mirhosseini et al. [30] investigated the use of TEGs for waste heat recovery in cement kilns with three different types of heat sinks and optimized TEGs for highest power output and lowest investment cost. The results revealed that the best cost per power ratio is attained with the staggered configured heat sink; however, different fill factors should be used at different sections of the kiln. There have been many other studies that explore the feasibility of the TE technology as a heat recovery system [31,32,33,34,35].

For waste heat recovery in gas turbines, Wu et al. [36] investigated a small TEG heat recovery unit on a gas turbine to power a sensor node. An experimental prototype, consisting of a TEG module and two heat pipes to dissipate heat from the cold side, was developed to characterize the performance and validate the mathematical model. At 325 °C heat source temperature, the system provided 0.92 W power output with a peak open circuit voltage of 2.4 V, which was enough to power few sensors and their auxiliary electronics. Bardy et al. [37] provided a mathematical model to predict the maximum efficiency of a TEG system integrated with a gas turbine system in two configurations: topping cycle configuration and preheating topping cycle configuration. In the former, TEG modules were placed on top of the gas turbine cycle between the heat source and the combustion chamber, whereas, in the later configuration, TEG modules received heat from the heat source to generate electricity and reject the remaining energy directly to the working fluid before it enters the combustion chamber. It was concluded that the topping configuration can improve the efficiency of the combined cycle and it was efficient for both low- and high-temperature Brayton cycles.

In this study, we investigate the “bottom cycle configuration”, where a thermoelectric waste heat recovery system (TEWHR) is mounted at the outer surface of the exhaust ducts in a gas turbine power plant to reduce the amount of wasted energy. We developed a numerical algorithm to optimize the TEWHR performance along with the calculation of exergy factors such as exergy efficiency, waste exergy ratio, and recoverable exergy. This numerical tool can solve the coupled heat and electric current equations with convective heat transfer at both sides, and account for the temperature reliance of the thermoelectric properties. Finally, considering that the thermal resistance of the cold side is one of the prominent factors that influence the performance of the system, the effect of the cold side convection heat transfer coefficient on the matched-load power output is investigated as well.

## 2. Methodology and Computational Models

### 2.1. Thermoelectric Modeling

Figure 1b shows a schematic of the gas turbine power plant with TEGs mounted at the exhaust duct of the turbine that we investigate. The compressor pressurizes the working fluid (air in this case) and then routes it to the combustion chamber, where heat energy is added to raise its temperature up to the design temperature limit of the turbine. After that, the pressurized hot gases expand in the prime mover and mechanical energy is produced. In this study, the LM6000 PC gas turbine (General Electric, Boston, MA, USA) of 42% thermal efficiency and 46 MW rated power output is used for investigation. This gas turbine is widely used in power plants and marine application to generate electricity and heat [38]. The flue gases leave the turbine at 500 °C, which falls within the range of high-quality heat, at a mass flow rate of 130 kg/s. The cross-sectional area and the length of the exhaust duct investigated are assumed to be 0.5 × 0.5 m^2^ and 2 m, respectively.

TEG modules are installed at the outer surface of the exhaust duct as shown in Figure 1b. Due to the temperature difference between the hot gases inside the duct and the surrounding air outside, heat flows through the attached TEGs by conduction and voltage is generated by the Seebeck effect. When a load resistance is connected to the module, electric current flows to the load by the generated voltage and thus electric power is delivered to the load. A detailed numerical algorithm has been developed in this study to investigate the power generation performance and to optimize the design of this thermoelectric waste heat recover system.

Our computational model incorporates the TE phenomena and the convective heat transfer phenomenon at both interfaces with flue gases and surrounding ambient air. For the TE phenomena (Seebeck, Peltier, Joule, and Thomson effects), the coupled thermal and electrical current equations are solved simultaneously in each TE element to find the temperature profile along its thickness, the rate of heat transfer through it, and the power output. A finite element method is used to divide the TE leg into smaller segments to obtain accurate temperature profile and to account for the change of material properties along the element thickness with temperature as shown in Figure 2a. Each segment has its own TE properties which correspond to the average temperature of the segment $T_{i}=\frac{T_{i}+T_{i+1}}{2}$. The expressions that describe the change of material properties with temperature have been curve fitted and integrated in the proposed algorithm. The three TE equations, conduction heat in Equation (2), Joule heat in Equation (3), and Peltier heat in Equation (4), are solved together in an iterative manner. The process starts with a linear temperature distribution inside the TE leg as an initial guess, assuming that the conduction heat dominates over Joule and Peltier heats. Before each iteration, the TE material properties are updated based on the temperature profile of the previous iteration. Hence, the fixed point iteration method has been used. In our model, we additionally assumed that (1) both heat and electrical transport along TE elements are 1D transport, (2) the inlet exhaust gas temperature and the ambient temperature are constant, (3) electrode and contact resistances are negligibly small compared to those of TE elements in the TE modules, (4) radiative heat transfer is negligibly small compared to conduction and convective heat transfer at the heat sinks, and (5) interface resistances between heat sinks and TE modules and between TE modules and the exhaust surface are negligibly small as well.
(2)$$Q_{cond,i}=K_{i}(T_{i-1}-T_{i})$$
(3)$$Q_{joule,i}=\frac{1}{2}I^{2}R_{i}+\frac{1}{2}I^{2}R_{i+1}$$
(4)$$Q_{Peltier,i}=\left(S_{i+1}-S_{i}\right)T_{i}I$$
where I is the electric current, $K_{i}=\frac{\kappa_{i}A_{leg}}{\Delta x_{i}}$ is the thermal conductance, and $R_{i}=\frac{\sigma_{i}A_{leg}}{∆x_{i}}$ is the electrical resistance of the i-th segment with thickness of $\Delta x_{i}$. Figure 2b depicts the equivalent thermal circuit model of one segment. Applying the first law of thermodynamics at the *i*-th node gives Equation (5)
(5)$$Q_{cond,i}+Q_{joule,i}=Q_{cond,i+1}+Q_{Peltier,i}$$
where the index *i* spans from 1 to *N* − 1. The first and the last segments of the TE element are in contact with the copper plates that convey the electric current between legs. Therefore, the first segment of the TE element has one component of Peltier heat and one component of Joule heat at the bottom interface, Equations (6) and (7), while the last the TE element has one component of Peltier heat and one component of Joule heat at the top interface, Equations (8) and (9).
(6)$$Q_{Peltier,0}=S_{1}T_{0}I$$
(7)$$Q_{joule,0}=\frac{1}{2}I^{2}R_{1}$$
(8)$$Q_{Peltier,N}=S_{N}T_{N}I$$
(9)$$Q_{joule,N}=\frac{1}{2}I^{2}R_{N}$$
where $T_{0}\;and\;T_{N}$ are influenced by the external thermal resistances.

Every TE generator has multiple pairs n-type and p-type legs element connected electrically in series and thermally in parallel. The ratio of the area covered by these TE pairs to the total area of the substrate is called the fill factor, which is expressed as
(10)$$FF=\frac{N_{pair}(A_{n}+A_{p})}{A_{total}}$$
where $N_{pair}$ is the number of TE pairs, $A_{n}$ is the cross-sectional area of the n-type element, and $A_{p}$ is the cross-sectional area of the p-type element. This factor along with the hot side heat transfer coefficient $h_{H}$ control the rate of heat flow into the first segment of the TE elements. Thus, the rate of heat input into the 0-th node is
(11)$$Q_{0}=\frac{1}{FF}A_{leg}h_{H}(T_{gas}-T_{0})$$
where the first term of this equation $\left(\frac{1}{FF}A_{leg}\right)$ represent the fractional area of the hot plate that provides the input heat for each TE leg. On the other end of the TE leg, the cold side heat transfer coefficient $h_{C}$ controls the heat output from the last segment of the TE element; therefore, the rate of heat conduction out of the *N*-th node is
(12)$$Q_{N}=\frac{1}{FF}A_{leg}h_{C}(T_{N}-T_{C})$$

So far, the mathematical formulation of the problem is not closed as there are N+3 unknowns, N+2 temperature points, and the electric current I, and only N+1 equations, i.e., two more equations are still needed. The closure of this problem lies in the fact that the total heat input to the TE modules is equal to the convection heat transferred from the flowing fluid to the surface of the exhaust duct and that the open circuit voltage of the TEG is equal to the sum of the individual voltage of each TE leg as they are connected electrically in series.
(13)$$Q_{conv,in}=\left(1-FF\right)A_{total}(T_{H}-T_{o})$$
(14)$$\sum_{1}^{N_{pairs}}\sum_{i}V_{OC,i}=I(\sum \left(R_{int}+2R_{elec.}+2R_{cont.}\right)+R_{L})$$
where $\left(1-F\right)A_{total}$ is the fractional area of the hot plate that is in contact with the TE legs, $R_{int}$ is the internal resistance of each leg, $R_{elec.}$ is the electrode resistance that connects between the legs, and $R_{cont.}$ is the contact resistance, which we assume is negligibly small compared to those of TE elements in this study. The open-circuit voltage of each segment $V_{OC,i}$ can be calculated from the Seebeck relation:(15)$$V_{oc,i}=S_{i}(T_{i}-T_{i-1})$$

After obtaining the required number of equations, these equations are solved simultaneously multiple times until the temperature profiles of both legs converge. The flow chart of the TE algorithm is shown in Figure 3, and the performance parameters can be calculated as follows:(16)$$P_{out}=IV_{oc}$$
(17)$$\eta_{TE}=\frac{P_{out}}{Q_{in}}$$To calculate the hot side convection coefficient $h_{H}$, one needs to use Equation (18) which corresponds to the Nusselt number of turbulent conduit flow [39].
(18)$$Nu=\frac{(\xi /8)RePr}{1.07+12.7{(\xi /8)}^{0.5}({Pr}^{2/3}-1)}$$
where $\xi ={\left(0.79\ln\left(Re\right)-1.64\right)}^{-2}$ assuming smooth duct, *Re* is the Reynolds number given by $\frac{uD_{H}}{\nu }$, and *Pr* is the Prandtl number given as $\frac{\nu }{\alpha }$. Numerous kinds of heat exchanger have been tested in such TEG systems such as straight fins [31] or pin fins [30] on the hot side, and thermosyphon [40] or flat plates on the cold side. Therefore, various values of $h_{H}$ and $h_{c}$ are investigated to evaluate their effects on the output power of the TEG system and the consequence of utilizing various heat exchangers. However, the system optimization has been first examined with typical values of a hot side convection coefficient of $1000\frac{W}{m^{2}\cdot K}$ and a cold side convection coefficient of $300\frac{W}{m^{2}\cdot K}$.

**Figure 3 entropy-25-01583-f003:**
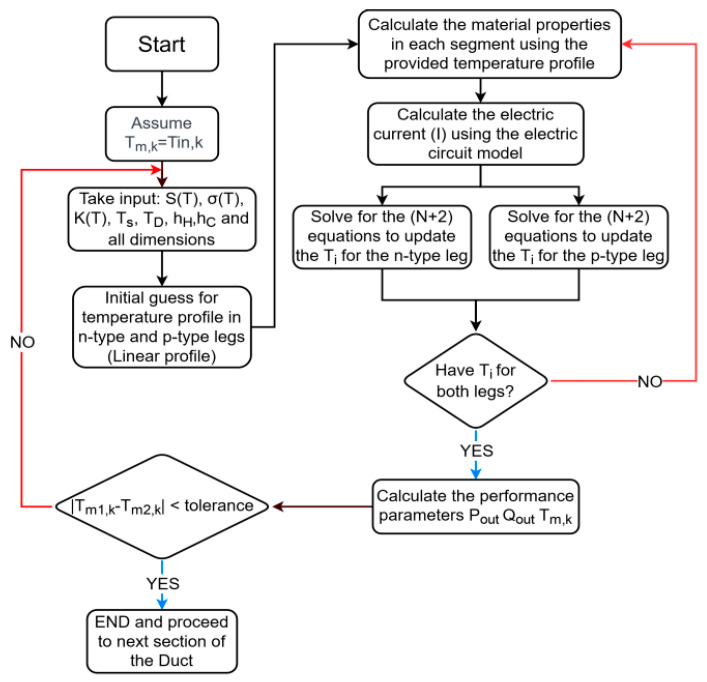
Flow chart of the combined heat transfer-TE simulation algorithm.

It is worth mentioning that the rate of heat transport through the n-type leg and the p-type leg are not equal because they have different thermal conductivities, and thus different temperature profiles along their thickness. Relying on this fact, Gosh et al. [21] argued that there is a lateral heat transport through the copper connecting plates, and, accordingly, they modified the heat input assigned to each leg to match the contact surface temperature. To investigate the viability of this argument, a numerical experiment has been carried out on a pair of TE elements using ANSYS-TE. The boundary conditions that have been used in this simulation are temperature based where the hot side temperature is set to 500 °C and the cold side temperature is set to 27 °C. The results reveal that the heat flux input to each leg comes from the top surface and from the middle section of the connecting plate as shown in Figure 4. Therefore, we conclude that there is no need to modify the heat input equations and make a two-level iterative algorithm.

The temperature of the released gases drops along the duct length as a result of heat transfer to the TE generators that are mounted on the outer surface of the duct; thus, an extensive heat transfer analysis needs to be carried out to find temperature distribution along the duct length. Therefore, one needs to divide the duct into n number of smaller sections to account for the influence of this temperature variation on the TE properties and the rate of power generation from the TEGs (Figure 5). Each section has a mean temperature $T_{m}^{k}$, which is the heat source temperature for the TEMs attached to this segment, equal to the average of the inlet and outlet mean temperatures of the *k*-th section, i.e., $T_{m}^{k}=\frac{T_{m}^{k,in}+T_{m}^{k,out}}{2}$. Always, in the first iteration of each segment, the mean temperature is assumed to be equal to the inlet temperature of that segment. Then, this is taken as the heat source temperature to start solving the TE equations which eventually provide the amount of heat absorb from this segment $Q_{k}$ among many other parameters.

Thereafter, the obtained value is plugged into Equation (19) to obtain the outlet temperature $T_{m}^{k,out}$, and the new mean temperature $T_{m}^{k}$ is calculated and compared with the one obtained from the previous iteration to check whether the convergence $\left(\left|T_{m}^{k,new}-T_{m}^{k,old}\right|\right)$ reach a stipulated tolerance or not. After reaching convergence, the solver proceeds to the next duct of the section to repeat the same process and find the accurate mean temperature. The flow chart of the combined algorithm is shown in Figure 3.
(19)$$Q_{k}=\dot{m}C_{p}(T_{m}^{k,in}-T_{m}^{k,out})$$

### 2.2. Exergy Analysis

Exergy is defined as a thermodynamic property that determines the amount of energy that can be extracted from an available energy source. Exergy analysis is a crucial step prior to the implementation of any energy conversion or heat recovery process as it helps to evaluate and optimize the process design. It can provide an accurate perception of how much the proposed design shifted from the ideal and identify the types and causes of irreversibility that yield to this shift. Furthermore, exergy can provide an insight into the environmental impact of the proposed heat recovery system and its sustainability through quantifying several exergy related factors such as the second-law efficiency, waste exergy ratio, exergetic improvement potential, and the recoverable exergy. All these factors are known as exergetic sustainability indicators, and they are defined as follows [38,41].

The second-law efficiency is the ratio of the useful exergy output to the exergy input of the TEG system such that
(20)$$\eta_{II}=\frac{Ex_{u}}{Ex_{input,TE}}$$
where $Ex_{u}$ is the useful exergy output from the TE module, and ${Ex}_{input,TE}$ is the exergy input to the TE module. From the second law perspective, the exergy efficiency represents the ratio of the actual power produced to the power that would have been produced when there are no thermodynamic irreversibilies in the system. In TE generation, the useful exergy output $\left(Ex_{u}\right)$ is equal to the power generated by the TEG module, while the exergy input (${Ex}_{input,TE}$) is equal to the heat exergy supplied from the flue gases. The input heat exergy equals the exergy difference across each section of the duct.

The waste exergy ratio (WER) is the ratio of the total wasted exergy $\left(Ex_{we}\right)$ to the total exergy input, given by
(21)$$WER=\frac{Ex_{we}}{Ex_{input,TE}}$$
where $Ex_{we}$ is the sum of the exergy destructed $\left(Ex_{dest.}\right)$ and the total exergy discarded to the environment $\left(Ex_{out,heat}\right)$, which is the heat exergy that leaves the TEG.

The exergy improvement potential, *IP*, is the prospective energy that could be extracted with further improvement of the TE generators, which could be done by enhancing the heat transfer coefficients across the TE legs or improving the figure of merit (*ZT*).
(22)$$IP=\frac{\left(1-\eta_{II})(Ex_{input,TE}-Ex_{u}\right)}{Ex_{input,TE}}$$It is noteworthy that these factors are evaluated for the TEG system separately. This means the mass exergy that leaves the system to the environment is not considered as waste exergy despite its huge amount.

The recoverable exergy is the potential exergy that could be extracted from the flue gas after it leaves the exhaust duct. It is equal to the difference between the exergy entering the exhaust duct and the exergy entering the TE generators from the flue gas. Thus, the recoverable exergy ratio, *RE*, is the ratio of the recoverable exergy to the exergy input to the exhaust duct.
(23)$$RE=\frac{Ex_{in}-Ex_{input,TE}}{Ex_{in}}$$

An extensive exergy analysis should be performed on the proposed system to be able to calculate the aforementioned exergy indicators. According to the laws of thermodynamics, exergy can be destroyed but cannot be created, and exergy destruction in any thermodynamic system is due to the presence of irreversibility inside this system, which can be sorted into two categories, namely internal irreversibility, and external irreversibility. In TE systems, the internal irreversibility factors include Joule heating, heat conduction in TE legs, and heat losses to the filler, while external irreversibility includes heat transfer with the heat source and heat sink, and fluid fiction. The exergy change of a system during a thermodynamic process is equal to the difference between the net exergy transfer and exergy destruction within the system, Equation (24).
(24)$$\dot{E}x_{in}-\dot{E}x_{out}-\dot{E}x_{dest}=\frac{dEx_{syst}}{dt}$$
where exergy can be transferred into or out of the system by heat, work, and mass flows across the system’s boundaries. Thus, the exergy of the *k*-th section of the duct can be written as the exergy of an open system in steady state operation, Figure 6.
(25)$$\dot{E}x_{dest}^{k}={\dot{E}x}_{mass,in}-{\dot{E}x}_{mass,out}-\dot{E}x_{heat,out}-\dot{E}x_{power,out}\;$$
where$\dot{E}x_{mass}=\dot{m}\left[C_{p}\left(T_{gas}-T_{0}\right)-T_{0}\left(s_{gas}-s_{0}\right)\right]$$\dot{E}x_{heat}=\left(1-\frac{T_{0}}{T_{c}}\right)Q_{rej}$$\dot{E}x_{power}=P_{TE,out}$$Q_{rej}$ is the heat rejected from the TE module. After calculating the exergy destruction of each section, the total exergy destruction of the TE heat recovery system is calculated using Equation (26).
(26)$$\dot{E}x_{dest,total}=\sum_{1}^{n}\dot{E}x_{dest}^{k}$$This exergy destruction comprises exergy destruction due to heat convection from the flue gases and exergy destruction due to heat transfer through the TEG modules, which can be calculated using Equations (27) and (28), respectively.
(27)$$\dot{E}x_{dest,conv}=\dot{m}\left[C_{p}\left(T_{in}-T_{out}\right)-T_{0}\left(s_{in}-s_{out}\right)\right]-\left(1-\frac{T_{0}}{T_{b}}\right)Q_{in}$$
(28)$$\dot{E}x_{dest,TE}=\left(1-\frac{T_{0}}{T_{b}}\right)Q_{in}-\left(1-\frac{T_{0}}{T_{C}}\right)Q_{rej}-P_{TE,out}$$
where $Q_{in}$ is the heat input to the TEG at the boundary temperature of the exhaust duct $T_{b}$.

**Figure 6 entropy-25-01583-f006:**
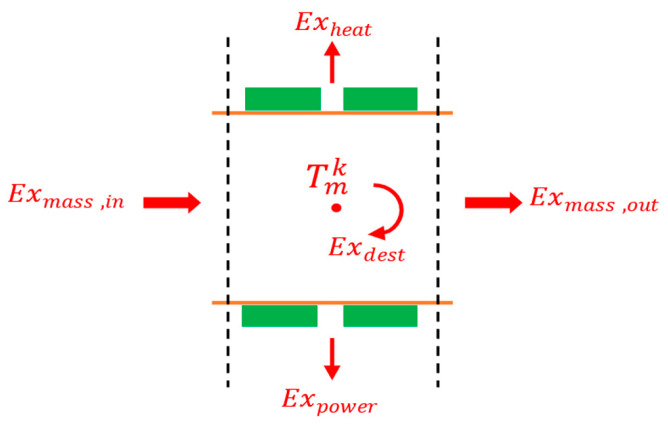
Exergy balance on one (*k*-th) section of the exhaust duct.

### 2.3. Economic Analysis

The viability of TE heat recovery systems must also be addressed economically to be able to compare it with other heat recovery technologies. The economic analysis of the proposed system in this work considers only the capital cost of the TE material. The capital cost (*CC*) incorporates the market cost of the TE material, the manufacturing cost of the TE modules, and the heat sink cost. However, this study considers only the raw volumetric TE material cost, Equation (29).
(29)$$CC=C_{TE,material}=\rho LFFA_{total}Y_{TE}$$
where ρ is the density of the TE material, L is the length of the TE elements, and $Y_{TE}$ is the unit mass cost of the TE material assumed to be 60 $/kg [42]. It is worth emphasizing that the capital cost and the performance of the systems are both functions of the filling ratio and the element length. Thus, optimization of both parameters could be carried out by evaluating the cost of a unit power for various FF and/or L (in case of non-matching load condition).

## 3. Results and Discussion

Bismuth telluride alloys have been known for long as the best TE materials within the specified temperature range [43]; therefore, they have been used in this study. The expressions that describe the change of material properties of Bi_0.5_Sb_1.5_Te_3_ (p-type) and Bi_2_Te_2.7_Se_0.3_ (n-type) with temperature have been curve fitted and integrated in the proposed algorithm [43,44,45].

### 3.1. Power Generation Performance

There are two independent design variables that can be adjusted to optimize the performance of a TE module, namely fill factor and TE leg thickness. Two output parameters can be monitored during optimization process, namely overall efficiency, and total output power. The total power output of the proposed heat recovery system is the sum of the power generated from each section of the exhaust duct, and its overall efficiency is the total power output divided by the total heat absorbed from the flue gases. The thermal and electrical resistances of a TEG module are functions of TE thickness and fill factor. When we vary the fill factor, we assume fixed cross-sectional areas for the TE legs, and change the number of legs in the module. As the fill factor increases, the thermal resistance decreases while the electrical resistance increases. As the leg thickness increases, both thermal and electrical resistances of the module increase. Hence, the performance of a TEG module is at its best when both types of resistances are matched with their corresponding external resistances [46]. Accordingly, the heat absorbed by the system increases with increasing fill factor (decreasing thermal resistance) and decreases with increasing length (increasing thermal resistance) as shown in Figure 7a, while the temperature difference across the TE legs steadily decreases with higher FF and increases with thicker legs is shown in Figure 7b. Moreover, owing to the large surface area with the installed TEG modules and the huge mass flow rate of the exhaust gas, the amount of heat input is on the order of $10^{5}$ W and the temperature difference across TE legs is higher than 200 K.

Figure 8a depicts the effects of fill factor and TE leg thickness on the total power output under the matched load condition. The figure reveals that the total power output of the system changes in a non-monotonic manner as fill factor increases from 0.1 to 0.9 at various leg lengths. For instance, at 3 mm and 6 mm leg thicknesses, the power output attains maximum values of 9.6 kW and 10.5 kW at FF=0.4 and FF=0.5, respectively, and then it starts to decrease. This non-monotonic change of power output is due to the conflicting effects between the fill factor and leg thickness as discussed above. Initially, increasing the number of TE legs, and thus increasing fill factor, leads to an increase in the amount of heat transport through the TE modules, Figure 7a. However, higher fill factor is not always in favor of increasing the power output because after a certain value, the temperature difference across the TE legs shown in Figure 7b starts to drop detrimentally which reduces the power output. Therefore, there is an optimal fill factor that maximizes power output for a fixed leg length as evidenced in Figure 8a. Moreover, it can be seen that a thicker TE leg requires higher fill factor for the maximum power output, this is because changing the leg length shifts the load matching conditions which push the optimum fill factor to compensate these changes and reaches larger values. A similar trend is observed when the length of the TE legs is changed; for instance, at the FF=0.4, the power output increases from 9.2 kW to 10.3 kW as the leg thickness increases from 3 to 6 mm, but it starts to deteriorate after 6 mm thickness. This is attributed to the rise of thermal resistance which induces the conflicting effect between the large temperature difference and the small heat transport rate; thus, the non-monotonic change with leg thickness is provoked. Moreover, one could easily notice that increasing the leg thickness to 12 mm or higher reduces the total power output of the system because this would increase the temperature difference across the TE legs which induce the bipolar transport and thus reduces the power factor.

The overall system efficiency is considered a combination parameter, as it is the result of the interaction between the total power output and total heat input. Thus, the change of this parameter should follow either the trend of one or the combined trend. For example, at a leg length of 3 mm, the overall efficiency keeps going down as the fill factor increases, Figure 8b. This means that at this leg length, the efficiency of the system is controlled by the amount of heat absorbed by the TEGs, which increases as FF increases. On the other hand, for leg lengths greater than 6 mm, the efficiency peaks at small fill factors and drops dramatically at high fill factors. At these length values, it is obvious that efficiency is influenced by both power and heat. Moreover, one could easily notice that the relationship between system efficiency and leg length is monotonically increasing which is attributed to the increase of thermal resistance that decreases the heat input to the TEGs, Equation (17).

The relationship of the cost per unit power with both fill factor and leg length is incrementally monotonic because the TE material cost is proportional to the volume of the material used. This is clearly manifested in Figure 9 over the studied ranges. Finally, the explicit goal that has been chosen for this optimization study is to achieve the highest possible power output at a reasonable capital cost. Thus, in Figure 8a, it can be noticed that the power increase is trivial after 6 mm thickness and cannot justify the extra economic price; therefore, 6 mm thickness and 0.5 fill factor are considered the optimum TEG input parameters for the proposed heat recovery system which can provide a total power output of 10.5 kW with 0.23 $/W. It is worth emphasizing that this cost includes the cost of the raw TE material only, and that the manufacturing and the heat sink cost will surge the capital cost of the heat recovery system.

### 3.2. Results of Exergy Analysis

The exergy flow from the flue gases to the outside environment is shown in Figure 10. The total exergy input to the TEG modules equals to 132.4 kW, which corresponds to 1% of total exergy available from the exhaust gases, at our optimal design with fill factor 50% and leg thickness 6 mm. About 5% of this exergy is destroyed due to the irreversibility associated with the convective heat transfer phenomenon at the hot side. Here, there are two sources of irreversibility: the heat transfer between fluid layers and fluid friction. The former mainly happens in the periphery layers where relatively steep temperature gradient is witnessed especially when the temperature profile is still developing, which is the presented case as we consider installing TEGs near the entrance of the duct. The small value of this irreversibility is attributed to the small temperature difference between the duct mean temperature and the boundary temperature. For example, for section number one, the mean temperature is 773.0 K while its boundary temperature is 718.8 K, and this temperature difference decreases along the duct length. The irreversibility due to fluid friction happens over the whole radial range, but is small compared to the heat transfer irreversibility. Regarding the TE exergy destruction and cold side exergy loss by heat, they make up about 30% and 57% of the exergy input, respectively. Finally, only about 8% of the input exergy is converted into useful power output which implies that there is considerable potential for improvement. This exergy efficiency could be improved by enhancing *ZT* of the used TE material.

The exergy values of the gas turbine components shown in Table 1 are obtained from [38], which performed an extensive exergy analysis to investigate the sustainability factors of a gas turbine power plant combined with Rankine cycle. Figure 11 shows the exergy efficiency of these components, in addition to the exergy efficiency of the gas turbine cycle and the proposed TE heat recovery system. The histogram manifest that the HPT and the LPT have the highest exergy efficiencies followed by the LPC and the HPC, 97%, 95%, 89%, and 87%, respectively, whereas the exergy efficiency of the combustion chamber is 85%, thanks to their high isentropic efficiencies [38]. Furthermore, the exergy efficiency of the gas turbine cycle is 39% and that of the proposed TE heat recovery system is 8%. Thus, the TE heat recovery system depicts the lowest exergy efficiency, which is attributed to the vast amount of exergy destruction as shown in Figure 12. This figure shows the variation of the TE exergy destruction and the convective exergy loss along the exhaust duct length. The linear trend of these plots is due to the linear temperature drop from 773.15 K at the inlet of the duct to 771.62 K at the exit. Obviously, convective exergy destruction is smaller than TE exergy destruction. This is because the temperature difference between fluid layers is relatively small, as discussed earlier, when compared to the temperature difference across the TE legs.

The exergy factors for the TE heat recovery system are presented in Figure 13. The recoverable exergy, RE, makes up 99% of the exergy that leaves the gas turbine which means only 1% of that exergy is utilized by the heat recovery process with our 2 m TE system. On the other hand, 92% of the exergy input to the TEG system is wasted, i.e., WER = 92%, as exergy destruction (35%) and as heat to the environment (57%) as was shown in Figure 10. From this perspective, future research will investigate the use of Ranking cycle for heat recovery from gas turbines with TE generators as an intermediate system between the two cycles where the heat rejected from the TE generators is exploited in the combined cycle instead of directly releasing it to the environment. As for the improvement potential (IP) of this heat recovery system, the exergy that could be retrieved from the exergy lost on the way from the flue gases to the environment through the TEG is found to be ~85% of the exergy input to the TEG.

### 3.3. Effects of Convective Heat Transfer Coefficients

Among the fundamental factors that impact the optimum performance of TE devices are the cold side and hot side convection coefficients. The effect of external thermal resistances on both of power output and unit power cost is investigated and reported in Figure 14a,b, respectively. Evidently, increasing the convection coefficients increases the maximum power output of the system which in turn decreases the cost of the unit power. However, the power gradient is more significant at low convection coefficients; this is to say that, when $h_{h}$ and $h_{c}$ are less than 1000 W/m^2^·K, the change of power output is remarkable, whereas the increase of power output is relatively smaller if they are higher than 1000 W/m^2^·K. The power output of a TE device is directly proportional to the temperature difference across its ends; thus, as the heat transfer coefficients increase on one side or on both sides, the temperature difference increases considerably until it reaches certain value where it starts to plateau because the overall thermal resistance approach the thermal matching condition. It is worth emphasizing that thermal matching condition is temperature dependent because the thermal conductance of the of the TE material is function of temperature.

## 4. Conclusions

This study investigated the feasibility of TE waste heat recovery systems installed on the exhaust of a gas turbine power plant to retrieve some of the released waste heat. A numerical tool was developed to couple the convective heat transfer phenomenon from the flue gases with the TE effects inside the TEG modules, and to account for the temperature-dependent TE material properties. Furthermore, an extensive exergy analysis was performed to evaluate some of the exergy indicators that provide an insight into the performance of the proposed system and directions for future improvement. Our results reveal that for the used TE materials, Bi_0.5_Sb_1.5_Te_3_ as p-type and Bi_2_Te_2.7_Se_0.3_ as n-type, the optimum leg thickness and fill factor are 6 mm and 0.5, respectively. This optimum design condition is found to produce a power output of 10.5 kW and power output density of 2.6 kW/m^2^ with a unit cost of 0.23 $/W at the load matching condition. Increasing the leg thickness may increase the power output slightly, but could require a larger fill factor, which is detrimental in the power cost due to the increased materials cost. Our exergy analysis shows that the produced power is achieved by only 1% of the exhaust gas exergy being exploited by the TE system. Nearly 92% of the exergy input to TE system is wasted, among which 57% is rejected from the TEG to the surrounding and 35% is destructed during the heat convection from the flue gases into TE modules and heat conduction inside the TE legs. Finally, increasing hot- and cold-side convection coefficients beyond 1000 W/m^2^·K results in a relatively small improvement of power output, which may not justify the extra cost added for the high-performance heat sinks. Our work is purely theoretical research that was conducted based on several assumptions as discussed in Section 2. Experimental verification and validation of the assumptions will be important future research.

## Figures and Tables

**Figure 1 entropy-25-01583-f001:**
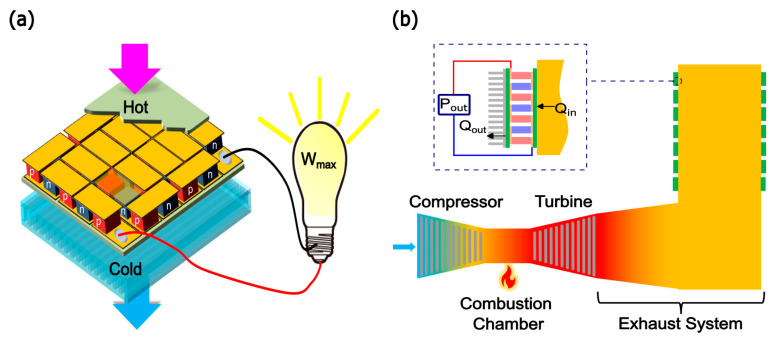
Schematic illustrations of (**a**) a thermoelectric generator (TEG) module and (**b**) a gas turbine cycle with TEG modules mounted at the exhaust duct.

**Figure 2 entropy-25-01583-f002:**
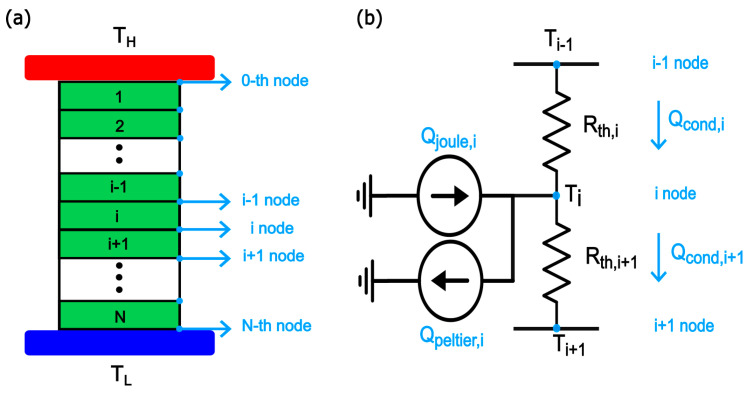
(**a**) Discretization of the TE element into N segments to account for the temperature variation along the length (**b**) Equivalent thermal circuit model of the *i*-th segment.

**Figure 4 entropy-25-01583-f004:**
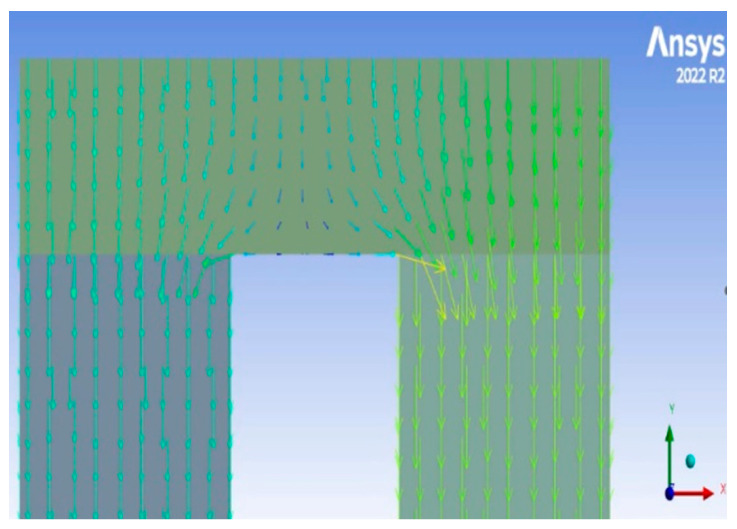
A 2D map of heat flux from a hot source at the top through a ceramic substrate and then through a pair of TE elements, showing the independent heat flow between the two TE elements and thus no lateral heat flow in the ceramic plate Note that this ANSYS simulation is conducted only to verify our assumptions that there is no significant lateral heat flow in the ceramic plate and the heat transport along TE elements is nearly 1D in TE elements. Our TE modeling does not use this ANSYS simulation.

**Figure 5 entropy-25-01583-f005:**
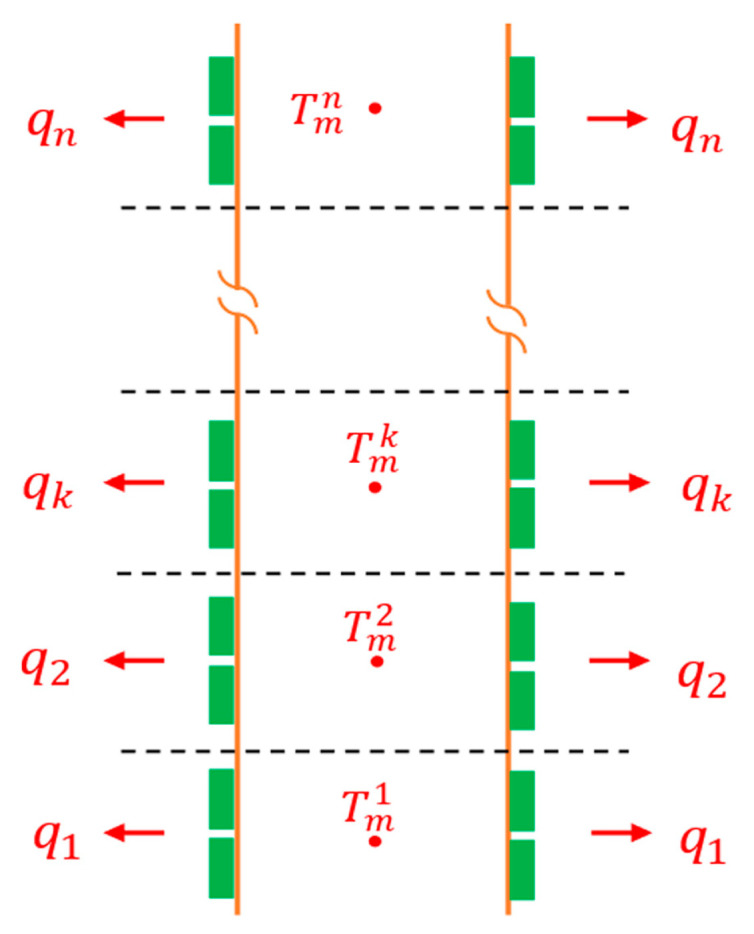
Discretization of exhaust duct to account for the temperature gradient along the length.

**Figure 7 entropy-25-01583-f007:**
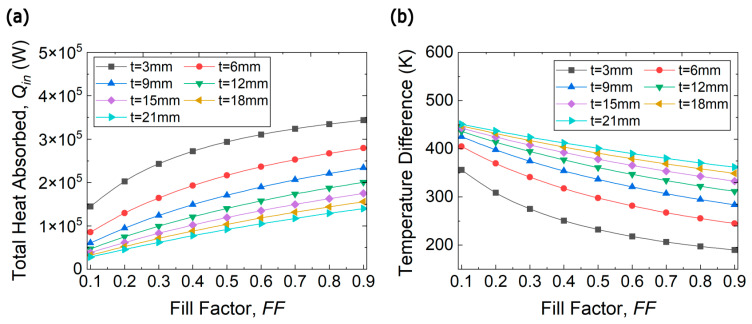
(**a**) Total heat input and (**b**) Temperature difference across TE legs for a waste heat recovery TE system mounted on the first section of the exhaust duct as a function of fill factor at various leg thicknesses.

**Figure 8 entropy-25-01583-f008:**
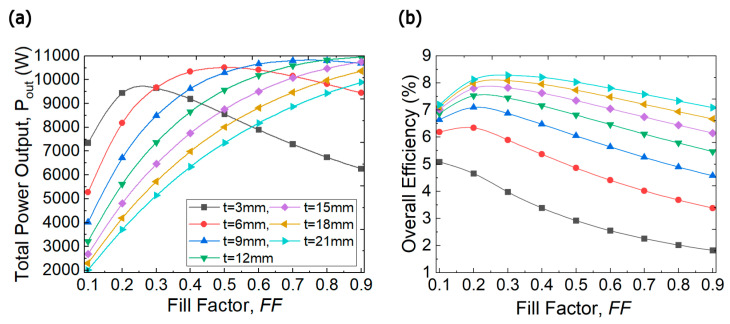
(**a**) Total power output and (**b**) Overall efficiency of the TEG heat recovery system as a function of fill factor at various leg thicknesses.

**Figure 9 entropy-25-01583-f009:**
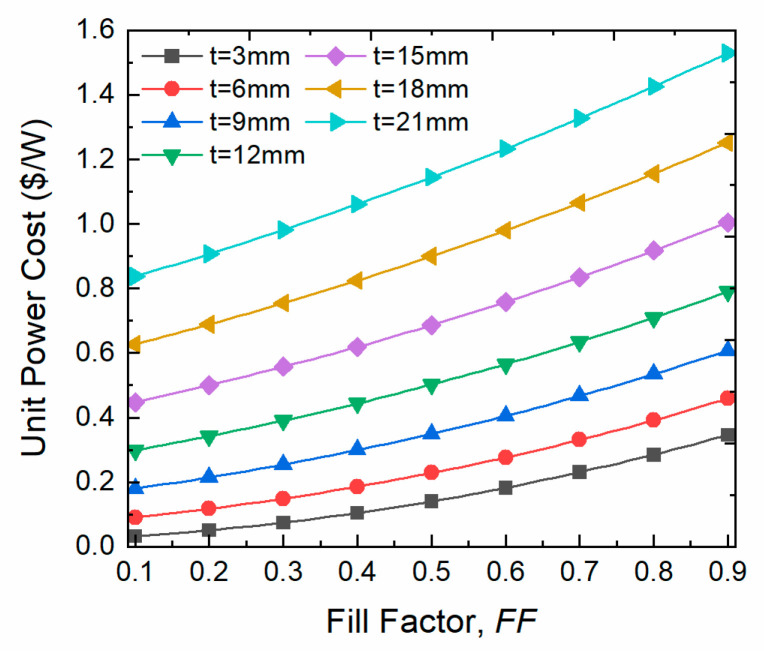
Power Cost of the TEG heat recovery system as a function of fill factor at various leg thicknesses.

**Figure 10 entropy-25-01583-f010:**
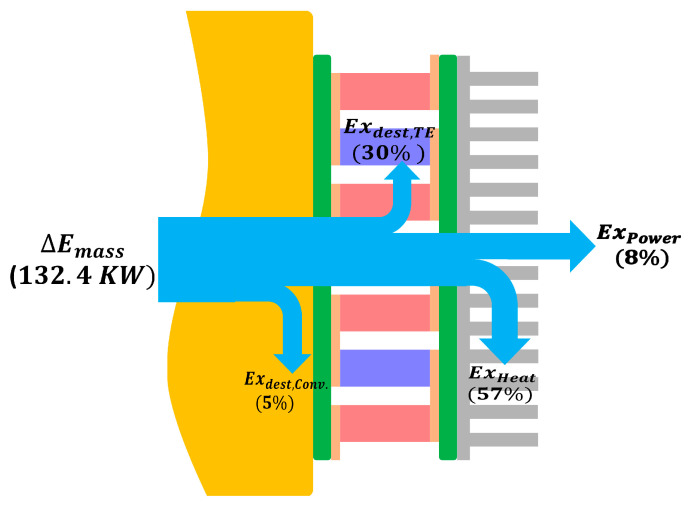
Exergy flow of the irreversible TE heat recovery system at the optimal design.

**Figure 11 entropy-25-01583-f011:**
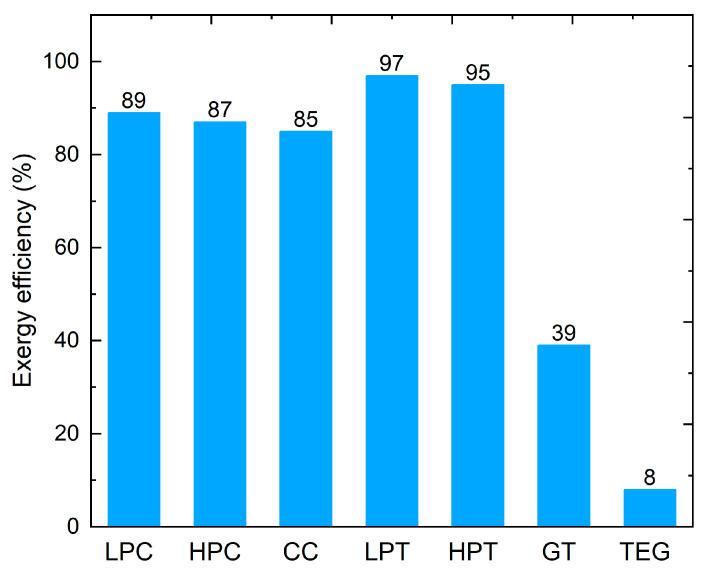
The second-Law exergy efficiency of the gas turbine components and the TE heat recovery system where LPC is low pressure compressors, HPC is high pressure compressors, CC is combustion chamber, LPT is low pressure turbine, HPT is high pressure turbine, GT is gas turbine cycle, and TEG is thermoelectric generators by this work.

**Figure 12 entropy-25-01583-f012:**
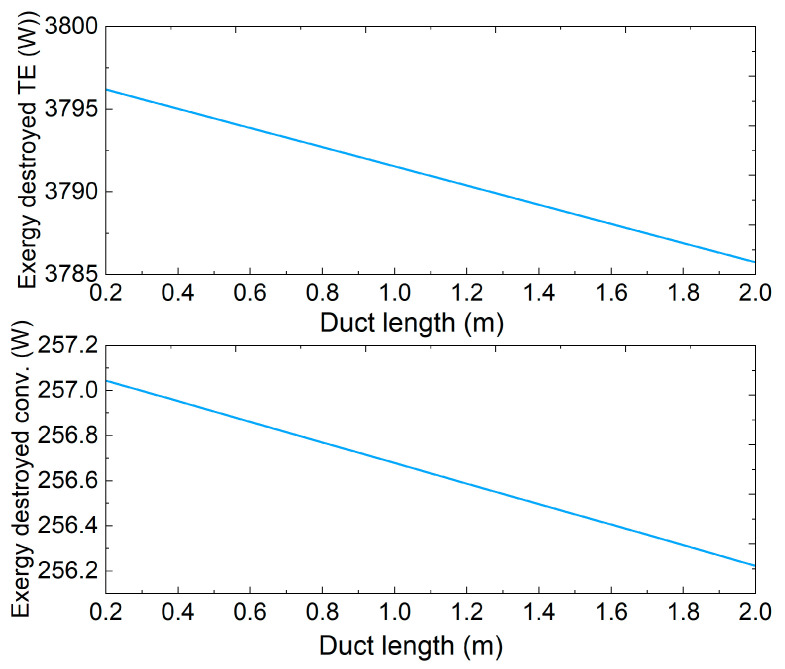
Convective exergy destruction which is due to heat transfer between fluid layers and fluid friction, and TE exergy destruction due to heat conduction in the p- and n-type legs. Both are along the duct length.

**Figure 13 entropy-25-01583-f013:**
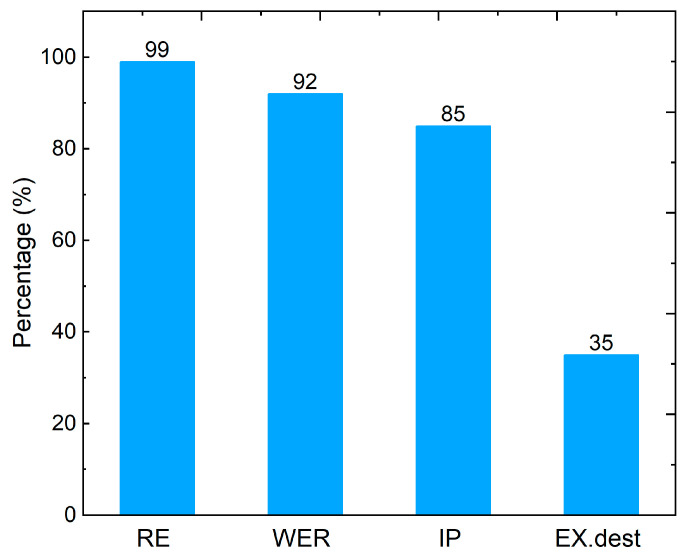
Exergetic sustainability indicators of the proposed TE heat recovery system, where RE is the recoverable exergy, WER is the waste exergy ratio, IP is the improvement potential, and EXdest is the exergy destroyed.

**Figure 14 entropy-25-01583-f014:**
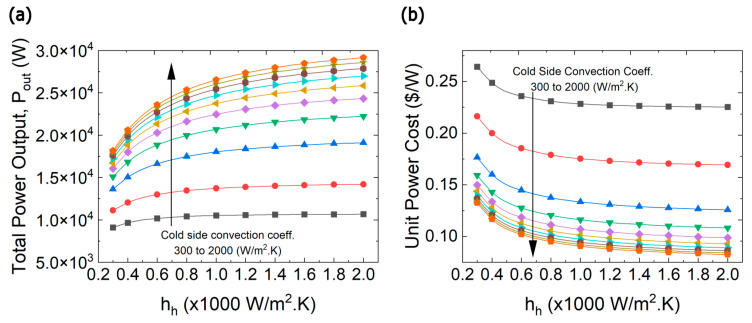
(**a**) Maximum power output, (**b**) Unit power cost as a function of convection coefficients. From black to orange curves as indicated by the arrow, the cold-side convection heat transfer coefficient was varied starting at 300 W/m^2^·K and then increasing from 400 to 2000 W/m^2^·K with a step size of 200 W/m^2^·K.

**Table 1 entropy-25-01583-t001:** Exergy efficiency of the components of gas turbine [37].

Components	Inlet Exergy (MW)	Outlet Exergy (MW)	Exergy Efficiency (%)
LPC	11.62	10.33	89
HPC	70.82	63.03	87
CC	174.18	148.27	85
HPT	148.27	146.65	97
LPT	85.56	82.84	95

## Data Availability

The data used in this study are available upon request from the corresponding author.

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
