# Peer review of "Performance Optimization and Exergy Analysis of Thermoelectric Heat Recovery System for Gas Turbine Power Plants"

_entropy, 2023, doi:10.3390/e25121583_

Round 1
Reviewer 1 Report
Comments and Suggestions for Authors
This work theoretically studies the performance optimization and exergy analysis of TE systems for recovering waste heat from the exhaust ducts of a gas turbine power plant. The topic is within the scope of this journal. Some improvements are still needed before publication.
1. Abstract--the research motivations should be better interpreted.
2. Introduction--the system-level applications of TE for photovoltaic/thermal or fuel cell system should be included. For example, potential evaluation of flexible annular thermoelectric generator in photovoltaic system performance improvement: energy and exergy perspectives.
3. Section 2--Model assumptions should be added. references supporting the formulas are also helpful.
4. Section 3--Please pay more attention on the scientific explainations of the results.
5. Conclusion section-- Please state the limitation or shortcomings of the study.
Comments on the Quality of English Language
Please avoid typo mistakes.
Author Response
We would first like to sincerely thank the reviewer for his/her very helpful comments on our manuscript. We have tried to address all the comments in our revised manuscript. Our detailed responses are shown below in this letter.
Reviewer #1
This work theoretically studies the performance optimization and exergy analysis of TE systems for recovering waste heat from the exhaust ducts of a gas turbine power plant. The topic is within the scope of this journal. Some improvements are still needed before publication.
Comment 1: Abstract--the research motivations should be better interpreted.
Response: Thank you for the suggestion. We have thoroughly revised Abstract to highlight the motivations behind this work including the following revised statement:
“…methods for application-specific, system-level TE design have not been thoroughly investigated. This work provides detailed design optimization strategies and exergy analysis for TE waste heat recovery systems. To this end, we propose the use of TE system equipped on the exhaust of a gas turbine power plant for waste heat recovery and use it as a case study. A numerical tool has been developed to solve the coupled charge and heat current equations with temperature-dependent material properties and convective heat transfer at the interfaces with the exhaust gases at the hot side and with the ambient air at the heat sink side…”
Comment 2: Introduction--the system-level applications of TE for photovoltaic/thermal or fuel cell system should be included. For example, potential evaluation of flexible annular thermoelectric generator in photovoltaic system performance improvement: energy and exergy perspectives.
Response: We have added the suggested paper, Lai et al., as Ref. [25], and also added a couple of sentences highlighting the results in Introduction as follows:
“…Lai et al. [25] investigated the viability of performance enhancement for a dye-sensitized solar cell coupled with selective solar absorber and flexible annular thermoelectric generator. The results revealed that the proposed system could produce 10.5 % higher power density and 39 % more maximum energy.”
Comment 3: Section 2--Model assumptions should be added. references supporting the formulas are also helpful.
Response: We have added the following lines in the end of the third paragraph of Sec. 2:
“In our model, we additionally assumed that (1) both heat and electrical transport along TE elements are 1D transport, (2) the inlet exhaust gas temperature and the ambient temperature are constant, (3) electrode and contact resistances are negligibly small compared to those of TE elements in the TE modules, (4) radiative heat transfer is negligibly small compared to conduction and convective heat transfer at the heat sinks, and (5) interface resistances between heat sinks and TE modules and between TE modules and the exhaust surface are negligibly small as well.”
We have already cited the source papers and books throughout Section 2.
Comment 4: Section 3--Please pay more attention on the scientific explanations of the results.
Response: We have added more interpretations related to heat transfer in the discussion section.
Comment 5: Conclusion section-- Please state the limitation or shortcomings of the study.
Response: Thank you for the suggestion. We have thoroughly revised Conclutions and discussed the limitations and assumptions made in our theoretical model, adding the following lines at the end of Conclusions:
“Our work is purely theoretical research conducted based on several assumptions as discussed in Sec. 2. Experimental verification and validation of the assumptions will be an important future work.”
Comment 6: Comments on the Quality of English Language: Please avoid typo mistakes.
Response: We have thoroughly checked the grammatical errors and typos throughout the manuscript and corrected/improved our English in the revised manuscript.
Thank you again for your helpful comments.

Reviewer 2 Report
Comments and Suggestions for Authors
1. The Abstract should be rephrased since it is hard to highlight more important information. Abstracts usually have at least one sentence per each: context and background, motivation, hypothesis, methods, results, conclusions.
2. The introduction must be reorganization. It is not clear the state of art in this field and the novelty of this reserch work and what's new in this article in the application to other articles.
3. What are the advantages and disadvantages of the thermoelectric generator
4. Figure 4 unclear, no scale, please describe the drawing in more detail, please describe what the drawing refers to
5. Figure 4 What phenomena does this figure describe and what effect does it have on the integration of thermoelectric modules with gas turbines?
6.No information about how the thermoelectric module is to be integrated into the gas turbine in the real solution.
7. Please indicate which boundary conditions were specified for the calculations in ANSYS-TE.
8. In order to assess the mesh quality, an analysis of the cell quality please add parameters such as :
- Orthogonal quality:
- Skewness
9. Line "The exergy values of the gas turbine components shown in Error! Reference source n 448
ot found. are obtained from [37]" please modify
10. Please elaborate more on your conclusions and refer to the results of your simulations in the article, It is not clear which is the significance of the results presented. results are not used for a proper discussion, evaluation of system performance or parametric analyses please modify.
Author Response
We would first like to sincerely thank the reviewer for his/her very helpful comments on our manuscript. We have tried to address all the comments in our revised manuscript. Our detailed responses are shown below in this letter.
Reviewer #2
Comment 1: The Abstract should be rephrased since it is hard to highlight more important information. Abstracts usually have at least one sentence per each: context and background, motivation, hypothesis, methods, results, conclusions.
Response: Thank you for the helpful comment. We have revised Abstract accordingly, clarifying the background and motivation behind this work. Now we believe that Abstract has all of these contents suggested by the reviewer.
Comment 2: The introduction must be reorganized. It is not clear the state of art in this field and the novelty of this research work and what's new in this article in the application to other articles.
Response: The introduction has been revised and fixed accordingly. We have extensive literature survey over 2 pages in Introduction to summarize the state-of-the-art technologies in TE waste heat recovery. The novelty of this work is two-fold as stated in the last paragraph of the section: (1) proposing a new bottom cycle configuration with TE modules attached at the exhaust side, and (2) formulating the relevant physical equations and algorithms to simulate realistic TE performance with temperature-dependent material properties and convective heat transfer at both sides.
Comment 3: What are the advantages and disadvantages of the thermoelectric generator
Response: Thank you for your question. This work is all about addressing this question in the context of waste heat recovery in gas turbines. The advantages are discussed in the fourth paragraph of Introduction in detail. The disadvantages or limitations are found from this work and discussed in Sec. 3 Results and Discussion, which include the low conversion efficiency and low exergy utilization. We have revised Conclusions accordingly, discussing the limitations.
Comment 4-5. Figure 4 unclear, no scale, please describe the drawing in more detail, please describe what the drawing refers to. Figure 4 What phenomena does this figure describe and what effect does it have on the integration of thermoelectric modules with gas turbines?
Response: The reason to add this figure is discussed in the last paragraph of Page 8. This figure represents the heat flux inside a pair of TE legs. It was presented to prove that there is no lateral heat flow in the ceramic plate to counterargue Ref. [21], and the heat transport is nearly 1D along the TE legs.
We have revised the corresponding text as well as the figure caption to make the argument clear:
“Note that this ANSYS simulation is conducted only to verify our assumptions that there is no sig-nificant lateral heat flow in the ceramic plate and the heat transport along TE elements is nearly 1D in TE elements. Our TE modeling does not use this ANSYS simulation.”
Comment 6: No information about how the thermoelectric module is to be integrated into the gas turbine in the real solution.
Response: We have stated in Introduction and Sec. 2 that our TEG modules are mounted on the external surface of the exhaust duct. The focus of this study is to investigate theoretically the viability of such systems in the botton cycle configuration. For the practical application, we may need to conduct further real-life investigations to find the best mounting options.
Comment 7: Please indicate which boundary conditions were specified for the calculations in ANSYS-TE.
We have added the following sentence about the boundary conditions in the last paragraph of Page 8:
“The boundary conditions that have been used in this simulation are temperature boundary conditions where the hot side temperature is set to 500℃ and the cold side temperature is set to 27℃.”
Please note as discussed above that this ANSYS simulation is only to verify the significance of lateral heat flow in the ceramic plate due to the different thermal characteristics of p-type and n-type TE elements. Our TE models do not use ANSYS, but our own codes in MATLAB.
Comment 8: In order to assess the mesh quality, an analysis of the cell quality please add parameters such as : Orthogonal quality, Skewness
Response: As discussed above, the ANSYS simulation was done separately from our TE modeling, only to verify the significance of lateral heat flow in the ceramic plate due to the different thermal characteristics of p-type and n-type TE elements. Our TE models do not use ANSYS, but we used our own codes in MATLAB. Therefore, the mesh quality has nothing to do with the quality of our TE modeling.
Comment 9: Line "The exergy values of the gas turbine components shown in Error! Reference source n 448 ot found. are obtained from [37]" please modify
Response: Thank you for the correction. The error has been fixed.
Comment 10: Please elaborate more on your conclusions and refer to the results of your simulations in the article, It is not clear which is the significance of the results presented. results are not used for a proper discussion, evaluation of system performance or parametric analyses please modify.
Response: We have improved Conclusions by clarifying the importance as well as the limitations of the results presented. We have mentioned that the proposed system could improve the power production by 10 KW and reduce the impact of waste heat on the environment. Moreover, we have shown the effects of many system parameters like the Fill Factors, the thickness of the TE modules, and the effect of convection coefficients.
Thank you again for your very helpful review comments.

Reviewer 3 Report
Comments and Suggestions for Authors
The manuscript falls comfortably within the scope of the paper. It diligently investigates the exergy analysis of a thermoelectric heat recovery system applied to gas turbine plants, with optimizing the system as a pivotal part of the study.
Here are my specific comments and recommendations for improvement:
- Development of a Computational Tool:
- The authors have articulated the development of a novel tool designed to solve coupled charge and heat current equations, considering temperature-dependent material properties and convective heat transfer at the interfaces. This is highlighted as the main contribution of the work. It would be highly beneficial if the authors could elaborate on this tool, providing more intricate details for reader comprehension.
- Additionally, I recommend that the authors consider making the codes pertinent to this study publicly available. Doing so would bolster the transparency and reproducibility of their findings and methodologies.
2. Quality of Illustrations:
- There’s a notable mention of Figure 3 in the manuscript. It’s advised that the authors revisit and enhance the quality and clarity of this figure slightly, ensuring that it effectively communicates the intended information and complements the text appropriately.
3. Discussion on Limitations:
- It would also be insightful if the authors could include a discussion on the potential limitations of their work. Such an inclusion would help provide a balanced perspective, allowing readers to gauge the applicability and reliability of the research findings.
Author Response
We would first like to sincerely thank the reviewer for his/her very helpful comments on our manuscript. We have tried to address all the comments in our revised manuscript. Our detailed responses are shown below in this letter.
Reviewer #3
The manuscript falls comfortably within the scope of the paper. It diligently investigates the exergy analysis of a thermoelectric heat recovery system applied to gas turbine plants, with optimizing the system as a pivotal part of the study.
Thank you for your favorable comments on our manuscript.
Here are my specific comments and recommendations for improvement:
Commet 1: Development of a Computational Tool:
- The authors have articulated the development of a novel tool designed to solve coupled charge and heat current equations, considering temperature-dependent material properties and convective heat transfer at the interfaces. This is highlighted as the main contribution of the work. It would be highly beneficial if the authors could elaborate on this tool, providing more intricate details for reader comprehension.
- Additionally, I recommend that the authors consider making the codes pertinent to this study publicly available. Doing so would bolster the transparency and reproducibility of their findings and methodologies.
Response: Thank you for the suggestion. We have articulated our modeling approach as detailed as it can be. The basics of thermoelectric conversion physics have been extensively discussed in literature since 1950s, and thus we have not explained the fundamentals in this manuscript. Instead, we have provided a step-by-step modeling method along with the iterative algorithm employed in this work to make sure that people can duplicate our work in the future.
Regarding the publication of the codes, we are planning to make a simulation tool and publish it on nanoHUB.org, so that people can run the same simulation for free of charge in the future. We have not decided yet though whether it is going to be an open-source or not. Nonetheless, we appreciate the reviewer’s insightful suggestion.
Comment 2: Quality of Illustrations:
- There’s a notable mention of Figure 3 in the manuscript. It’s advised that the authors revisit and enhance the quality and clarity of this figure slightly, ensuring that it effectively communicates the intended information and complements the text appropriately.
Response: Thank you for your suggestion. The figure quality and readability has been improved and fixed.
Comment 3: Discussion on Limitations:
- It would also be insightful if the authors could include a discussion on the potential limitations of their work. Such an inclusion would help provide a balanced perspective, allowing readers to gauge the applicability and reliability of the research findings.
Response: Thank you for the suggestion. We have added sentences in Conclusions about the limitations of our approach:
“Our work is purely theoretical research conducted based on several assumptions as discussed in Sec. 2. Experimental verification and validation of the assumptions will be an important future work.”
Thank you again for your helpful comments and suggestions. We appreciate it.

Round 2
Reviewer 1 Report
Comments and Suggestions for Authors
This paper has been greatly improved according to the reviewer's comments.
Reviewer 3 Report
Comments and Suggestions for Authors
The authors appropriately revised the previous version of the manuscript.